# Molecular Analysis of Biliary Tract Cancers with the Custom 3′ RACE-Based NGS Panel

**DOI:** 10.3390/diagnostics13203168

**Published:** 2023-10-10

**Authors:** Natalia V. Mitiushkina, Vladislav I. Tiurin, Aleksandra A. Anuskina, Natalia A. Bordovskaya, Anna D. Shestakova, Aleksandr S. Martianov, Mikhail G. Bubnov, Anna S. Shishkina, Maria V. Semina, Aleksandr A. Romanko, Ekaterina S. Kuligina, Evgeny N. Imyanitov

**Affiliations:** 1Department of Tumor Growth Biology, N.N. Petrov Institute of Oncology, 197758 St. Petersburg, Russia; nmmail@inbox.ru (N.V.M.); bordovskaya.n11@gmail.com (N.A.B.); pluto8ho@gmail.com (A.S.S.); mvsyomina@gmail.com (M.V.S.); romanko.aleksandr.a@gmail.com (A.A.R.); kate.kuligina@gmail.com (E.S.K.); 2Department of Medical Genetics, St.-Petersburg Pediatric Medical University, 194100 St. Petersburg, Russia

**Keywords:** biliary tract cancer, cholangiocarcinoma, targeted RNA sequencing, 3′ RACE, *FGFR2* translocation, NGS library preparation

## Abstract

The technique 3’ rapid amplification of cDNA ends (3′ RACE) allows for detection of translocations with unknown gene partners located at the 3′ end of the chimeric transcript. We composed a 3′ RACE-based RNA sequencing panel for the analysis of *FGFR1–4* gene rearrangements, detection of activating mutations located within *FGFR1–4*, *IDH1/2, ERBB2 (HER2), KRAS, NRAS, BRAF,* and *PIK3CA* genes, and measurement of the expression of *ERBB2, PD-L1,* and *FGFR1–4* transcripts. This NGS panel was utilized for the molecular profiling of 168 biliary tract carcinomas (BTCs), including 83 intrahepatic cholangiocarcinomas (iCCAs), 44 extrahepatic cholangiocarcinomas (eCCAs), and 41 gallbladder adenocarcinomas (GBAs). The NGS failure rate was 3/168 (1.8%). iCCAs, but not other categories of BTCs, were characterized by frequent *FGFR2* alterations (17/82, 20.7%) and *IDH1/2* mutations (23/82, 28%). Other potentially druggable events included *ERBB2* amplifications or mutations (7/165, 4.2% of all successfully analyzed BTCs) and *BRAF* p.V600E mutations (3/165, 1.8%). In addition to NGS, we analyzed microsatellite instability (MSI) using the standard five markers and revealed this event in 3/158 (1.9%) BTCs. There were no instances of *ALK, ROS1, RET,* and *NTRK1–3* gene rearrangements or *MET* exon 14 skipping mutations. Parallel analysis of 47 iCCA samples with the Illumina TruSight Tumor 170 kit confirmed good performance of our NGS panel. In conclusion, targeted RNA sequencing coupled with the 3′ RACE technology is an efficient tool for the molecular diagnostics of BTCs.

## 1. Introduction

Biliary tract cancers (BTCs) are classified as intrahepatic cholangiocarcinomas (iCCAs), extrahepatic cholangiocarcinomas (eCCAs), and gallbladder adenocarcinoma (GBAs). The incidence of BTCs is approximately 2–3 per 100,000 people in most countries; however, it is significantly higher in geographic regions where liver flukes are endemic [1,2]. The prognosis for patients with BTCs is poor, with 5-year overall survival rates falling below 20% [2,3]. BTCs are commonly diagnosed at an advanced stage and require systemic therapy. The combination of gemcitabine and cisplatin was the most common first-line treatment option before the invention of immune therapy. Recent clinical trials have demonstrated that the addition of immune checkpoints inhibitors to this scheme may produce a significant survival benefit for a subset of patients; therefore, the combination of gemcitabine, cisplatin, and durvalumab has been accepted as a novel standard of care for the upfront treatment of advanced BTC [3,4]. 

The molecular characterization of BTCs has led to the development of new targeted therapies in the past few years. Three FGFR inhibitors, pemigatinib, infigratinib, and futibatinib, have been approved for the treatment of CCAs carrying *FGFR2* translocations [3,5,6,7]. Ivosidenib is an inhibitor of mutated isocitrate dehydrogenase-1 (*IDH1*) that is approved for the treatment of advanced CCA with *IDH1 c.132* mutations [8]. Translocations involving *FGFR2* gene and mutations in *IDH1* gene are among the most frequent genetic alterations detected in iCCAs. Other targetable alterations with a significant prevalence in BTCs include activating events in the *ERBB2 (HER2)* oncogene, *BRAF* p.V600E mutations, and *KRAS* p.G12C substitutions [3,9]. Thus, the selection of the best treatment options for patients with BTC requires complex molecular analysis of the tumor tissues. 

The detection of the *FGFR2* translocations is particularly challenging because such rearrangements can engage numerous partner genes. At the same time, the part of *FGFR2* gene that participates in translocations is conserved: the breakpoint is usually located after exon 17 or, less frequently, at the beginning of exon 18. Jusakul et al. [10], followed by Zingg et al. [11], demonstrated that the loss of the 3′ UTR region of the *FGFR2* gene itself leads to its activation, and such loss can result from different molecular events, such as diverse rearrangements, partial amplifications, protein-truncating, or spice-site mutations, etc. Importantly, some activating amino acid substitutions in the *FGFR2* gene can be targeted with FGFR inhibitors, similarly to the *FGFR2*-rearranged tumors [12,13].

Given high diversity of *FGFR2* gene alterations in cholangiocarcinoma, NGS can be regarded as an optimal diagnostic approach. Moreover, RNA sequencing should be preferred over DNA sequencing for the studies of oncogenic fusions. Indeed, at the DNA level, many translocations involve repeats or other intronic sequences that pose difficulties for the short-read sequence aligners. Furthermore, non-functional rearrangements with no or low expression of the product are sometimes detected via DNA sequencing (or FISH), whereas no such transcripts would be found in RNA sequencing data [14,15,16]. RNA sequencing can be used both for the detection of activating mutations and for the analysis of gene expression. Several approaches have been utilized for the construction of targeted RNA sequencing libraries, including hybridization-based, anchored multiplex PCR, and amplicon-based methods, all having their advantages and disadvantages [17,18,19,20]. The amplicon-based approach is fast and straightforward but can only identify known translocation variants, whereas target enrichment by hybridization is the most informative but also the most laborious and time-consuming method [17,18,19,20]. 

Anchored multiplex PCR methods are able to detect translocation with an unknown partner gene only if the breakpoint positions in at least one of the genes participating in the translocation are well defined [17]. They are generally more rapid and less expensive when compared with hybridization-based NGS library preparation protocols. PCR enrichment takes only about 2 h, whereas hybridization requires overnight incubation. Anchored multiplex PCR panels usually focus on a relatively narrow number of genomic segments and, therefore, they are smaller than hybridization-based NGS tests. Additionally, fewer NGS reads are required to achieve sufficient coverage in each region of interest and more samples can be multiplexed for each run when PCR enrichment is utilized. The most commonly used anchored multiplex PCR methods require the RNA to be converted to double-strand cDNA, followed by the adapter ligation, PCR enrichment, and final amplification of the libraries [17]. On the basis of this approach, two popular commercial NGS solutions were developed, namely Archer FusionPlex (ArcherDX, Boulder, CO, USA) and QIAseq RNAscan (Qiagen, Hilden, Germany). Multiple studies have relied on the use of these kits, thus supporting the feasibility of the anchored multiplex PCR method [21,22,23,24]. Alternatively, the adapter sequence can be attached to the cDNA molecules by the reverse transcriptase; this procedure is utilized in the template-switch-based 5′ rapid amplification of cDNA ends (5′ RACE) or the 3′ rapid amplification of cDNA ends (3′ RACE) protocols [25,26]. The 5′ RACE technique can be used when all the known translocation partners are located at the 3′ part of the chimeric transcripts, whereas the respective unknown partners are located at the 5′ part of the chimeric transcripts. On the other hand, 3′ RACE can be utilized only if all the unknown translocation partners are expected to occupy the 3′ ends of the mRNA. Importantly, the 3′ RACE- and 5′ RACE-based approaches allow for simple and straightforward in-house design of the custom RNA NGS tests.

Here, we developed an RNA sequencing panel for the detection of the main targetable alterations in BTCs on the basis of 3′ RACE technology and used it for the analysis of 168 clinical BTC samples.

## 2. Materials and Methods

### 2.1. Patients and Samples

The study included BTCs, which were referred for molecular testing to the N.N. Petrov Institute of Oncology (St. Petersburg, Russia) in the years 2022–2023. The inclusion criteria were as follows: (1) diagnosis of BTC, which could further be classified as iCCA, eCCA, or GBA, (2) presence of FFPE material with at least 5% tumor cell content, and (3) patient’s written informed consent for participation in the molecular genetic study. The study recruited 168 patients: 83 patients with iCCA, 44 patients with eCCA (including 10 patients with perihilar CCA), and 41 patients with GBA. The investigation was conducted in accordance with the Helsinki Declaration and was approved by the local ethics committee. Total nucleic acids from FFPE tissues were extracted using a Quick-DNA/RNA MagBead kit (Zymo Research, Irvine, CA, USA). *ERBB2* amplifications and MSI status (standard pentaplex panel: *BAT25*, *BAT26*, *NR21*, *NR22,* and *NR24* mononucleotide markers) were determined as described previously [27]. For the iCCA samples that were not subjected to the NGS analysis using Illumina TruSight Tumor 170 kit, receptor tyrosine kinase rearrangements (*ALK, RET, ROS1, RET,* and *NTRK1–3*) and *MET* exon 14 skipping mutations were analyzed via PCR tests as described previously [28,29,30,31]. 

### 2.2. Preparation of NGS Libraries Using 3′ RACE-Based Method

RNA content was assessed in each sample by NanoPhotometer (Implen GmbH, München, Germany). The concentrations within the range 50–500 ng/μL were considered suitable for the library preparation. Samples with lower nucleic acid contents were concentrated using vacuum centrifuge at 60 °C from the volume of 40–50 μL to 10.5 μL, and the full sample was used in a single analysis. Samples with extremely high concentrations (>500 ng/μL) were found to have an elevated DNA to RNA ratio; these samples required DNase pretreatment prior to the reverse transcription reaction. For this purpose, we measured RNA content using a Qubit 4.0 fluorometer (Thermo Fisher Scientific, Waltham, MA, USA) and took 1 μg RNA in the reaction with DNase I (Amplification Grade, Invitrogen, Waltham, MA, USA), which was carried out according to manufacturer’s instructions.

The library preparation protocol consisted of the three main steps: (1) reverse transcription reaction, (2) target enrichment by PCR, and (3) library amplification. 

Reverse transcription reaction included 10.5 μL of the RNA sample, 1 μL of RevertAid Reverse Transcriptase (Thermo Fisher Scientific, USA; 200U/μL), 4 μL of 5× Reaction buffer, 2 μL of dNTP mix (10 mM each), 2 μL of R10_AD2 primer (5′-GTTCAGACGTGTGCTCTTCCGATCTNNNNNNNNNN-3′, where N stands for the random nucleotide; 50 μM), and 0.5 μL of Thermo Scientific RiboLock RNase inhibitor (20 U/μL). First, all components of the reaction, except RevertAid Reverse Transcriptase and RiboLock RNase inhibitor, were mixed together, and the mixture was heated to 70 °C for 3 min to denature the RNA. After a short cooling period (3 min at room temperature and 3 min at 4 °C), the reaction was supplemented with the RevertAid Reverse Transcriptase and RiboLock RNase inhibitor and placed in the thermocycler with the following program: 25 °C for 10 min, 42 °C for 45 min, 45 °C for 15 min, and 70 °C for 10 min. After completion of the reaction, 20 μL of clean water was added and the cDNA was cleaned using 1.4× volume of AmPure magnetic beads (Beckman Coulter, Brea, CA, USA). 

The PCR-enrichment reaction included 25 μL of the cleaned cDNA from the previous step, 5 μL of 10× GeneAmp PCR Buffer I (Applied Biosystems, Waltham, MA, USA), 0.5 μL of TaqM polymerase (AlkorBio, St. Petersburg, Russia; 5 U/μL), 2 μL of MgCl2 (25 μM), 1.25 μL of dNTP mix (10 mM each), 2 μL of Primer Mix for Enrichment (0.5 μM of each primer), 2 μL of index primer I(N)-AD7 (5′-CAAGCAGAAGACGGCATACGAGATXXXXXXXXGTGACTGGAGTTCAGACGTGTGCTCTTCC-3′, where (N) stands for the index number and XXXXXXXX stands for the index sequences, different for each sample within a single run; 5 μM), and 12.25 μL of clean water. The PCR was carried out on the T100 thermal cycle instrument (Bio-Rad, Hercules, CA, USA). The reaction was started by the enzyme activation (95 °C, 10 min) and included 8 cycles (95 °C for 20 s, 63 °C for 10 min), followed by 5 min incubation at 72 °C. After the completion of the last step, the reaction was immediately placed on ice and cleaned with 1.2× volume of AmPure magnetic beads. 

Normally, the specificity of the PCR reaction is ensured by the use of two closely located primers that determine the unique region in the studied genome or transcriptome. However, anchored PCR methods (including 3′ RACE) require each single primer to possess enough specificity: ideally, it should be able to hybridize with just one genomic locus at the selected temperature. Recognizing the importance of the primer design, we developed the framework shown in Appendix A, which was used in the current study and may help in developing targeted RNA panels on the basis of anchored multiplex PCR approaches in the future. The full list of primer sequences composing the Primer Mix for the Enrichment in the current study can be found in Appendix A.

For the final amplification step, 25 μL of the cleaned enriched library was mixed with 5 μL of 10× GeneAmp PCR Buffer I, 0.5 μL of TaqM polymerase (5 U/μL), 2 μL of MgCl2 (25 μM), 1.25 μL of dNTP mix (10 mM each), 2 μL of the primer AMP-PCR2s (5′-CAAGCAGAAGACGGCATACG-3′; 5 μM), 2 μL of the index primer I(N)-AD5 (5′-AATGATACGGCGACCACCGAGATCTACACXXXXXXXXACACTCTTTCCCTACACGACGCTCTTCC-3′, where (N) stands for the index number and XXXXXXXX stands for the index sequence, which were different for each sample within a single run; 5 μM), and 12.25 μL of clean water. The reaction program consisted of the enzyme activation step (95 °C, 10 min), 28 amplification cycles (95 °C for 15 s, 60 °C for 30 s, 72 °C for 30 s), followed by a final elongation step (72 °C, 5 min). The reaction was placed on ice and then cleaned with 0.9× volume of AmPure magnetic beads.

The prepared libraries were measured with the Qubit 4.0 fluorometer and pooled according to their concentrations. The library size, determined for the pool of libraries using the 5200 Fragment Analyzer System (Agilent Technologies, Santa Clara, CA, USA), was about 250 bp for all prepared libraries. This size was used to dilute the libraries to a 2 nM concentration, as was required for NGS sequencing. The prepared pooled libraries were sequenced on Illumina MiSeq or NextSeq 550 instruments in paired-end mode (2 × 150 bp). We considered 100,000 pairs of reads to be enough for the analysis of one sample. 

### 2.3. Preparation of NGS Libraries Using the Illumina TruSight Tumor 170 Kit

RNA and DNA libraries were prepared from each of the selected samples according to the manufacturer’s instructions with the following exceptions. First, an enzymatic fragmentation was used for the preparation of DNA libraries instead of ultrasonication. The nucleic acid sample (up to 1 μg DNA, as measured by NanoPhotometer) was cut with the Nuclease P1 (New England BioLabs, Ipswich, MA, USA) as described previously [32], then cleaned with 1.8× AmPure magnetic beads. After the analysis of the fragment size distribution by the 5200 Fragment Analyzer System (Agilent Technologies, USA), if necessary, the sample was additionally cut for 5 min with the NEBNext dsDNA Fragmentase (New England BioLabs, Ipswich, MA, USA). The fragmented DNA (20–100 ng) was then subjected to the library preparation, starting from the end repair and A-tailing step. Second, instead of bead-based normalization, all prepared libraries were measured with the Qubit 4.0 fluorometer and the 5200 Fragment Analyzer System and pooled at equimolar concentrations for sequencing on the Illumina NextSeq 550 instrument.

### 2.4. Bioinformatics and Statistical Analysis

The bioinformatics pipeline, which involves the analysis of NGS 3′ RACE data and allows both the detection of translocations and mutations and the measurement of gene expression, is described in detail in Appendix A. Briefly, STAR-Fusion software v1.10.1 was used to detect rearrangements of the *FGFR1–4* genes in all BTC samples [33]. Unfortunately, the software developed for somatic mutation calling usually requires sequencing of the normal paired sample together with the tumor sample. We found that the available instruments that can be used in tumor-only mode often miss low-prevalent somatic variants. Because the sensitivity of a method used in a diagnostic procedure is of high importance, we developed custom scripts, primarily based on the *pysam* python module [34], which were utilized for filtering improperly aligned reads and variant calling. All scripts involved in this study are available from GitHub [https://github.com/MitiushkinaNV/RACE_NGS (accessed on 6 September 2023)]. The use of primers for the PCR-enrichment of targeted NGS libraries leads to poor discrimination of PCR duplicates, which has to be performed by using only a single coordinate. This can lead to underestimation of the coverage in some of the analyzed regions. To improve the analysis of regions with higher coverage, we decided to use the mismatches existing between the random primer used in the reverse transcription reaction and the actual mRNA sequence as pseudo-UMIs to distinguish different templates with the same genomic coordinates. The UMI-tools v1.1.4 package was used for this purpose [35]. The reads were aligned with the STAR program v2.7.8a [36]. The analysis of gene expression was performed in a qPCR-like manner, which is further explained in Appendix A. The expression of the *FGFR1* exons 17–18, *FGFR2* exons 16–17, *FGFR3* exons 14–15, *FGFR4* exons 14–15, *ERBB2* exons 19–20, and *CD274* (*PD-L1*) exons 5–6 was measured against the three referee genes, *DDX23* exons 8–9, *GOLGA5* exons 5–6, and *SEL1L* exons 13–14. The median of three normalized values was taken as a result for each of the genes of interest. The referee genes were selected based on the study of Eisenberg and Levanon [37]. For the three cases with less than 3000 templates per sample remaining after the removal of PCR duplicates, the results of the NGS analysis were interpreted as a failure. Generally, it proved difficult to select the proper threshold, which would indicate the success or failure of the analysis. Although it is reasonably likely that in samples with smaller numbers of analyzed templates or with a low coverage of the referee genes, the pathogenic mutations can be missed, we did not find clear dependence between these factors and the probability to detect mutations in the analyzed samples. Thus, the arbitrary threshold was selected. 

The analysis of data obtained with the Illumina TruSight Tumor 170 panel relied on the use of the TruSight Tumor 170 BaseSpace app [38]. The genomic variants outputted by the Illumina software v2.0.2 were annotated using ANNOVAR [39], as were the variants found with the 3′ RACE-based NGS. The filtering of the called genomic variants was performed in a similar way for both panels (Appendix A), although lower thresholds could be applied to the DNA sequencing results due to the lower error rate. All annotations of genetic variants provided in this study were performed using the Ensembl canonical transcripts only [40]. 

Statistical analysis of bioclinical associations was performed by the R software v4.1.1 [41]; the functions of the R *graphics* package were used for plotting.

## 3. Results

### 3.1. Development of the Targeted RNA Sequencing Panel on the Basis of 3′ RACE Technology

The scheme of the proposed method is given in Figure 1. The process of NGS library preparation starts with the reverse transcription reaction, in which tailed random primers are used to generate the adapter sequence at the 5′-end of each cDNA molecule. The next step is the target enrichment by PCR. A mix of primers, specific to the selected regions of interest, is used in the reaction together with a reverse primer, which anneals to the 5′ end adapter sequence. The latter contains one of the two index sequences required for the multiplexing of samples. During the last step, PCR amplification of the enriched libraries is coupled with the addition of the second index sequence. 

The composition of the targeted RNA panel developed for the study of the main clinically relevant genetic alterations in BTCs is presented in the Table 1. 

### 3.2. Analysis of BTCs Using the 3′ RACE Targeted RNA Sequencing Panel

Targeted RNA sequencing was performed for 168 BTC samples, including 83 iCCAs, 44 eCCAs, and 41 GBAs. The main clinical characteristics of the studied groups together with the results of the PCR and NGS analyses are summarized in Table 2. Three samples with poor RNA content failed the NGS analysis. The total number of templates per sample, calculated after removing PCR duplicates and reads lying outside of the specified coordinates, varied from 3393 to 69,133, with a median of 28,241 templates per sample.

Translocations involving the *FGFR2* gene were detected in 14 samples: *FGFR2-AHCYL1 (F17;A2)*, *FGFR2-BICC1 (F17;B3)* and *FGFR2-VCL (F17;V2)* variants were found in two cases each, whereas *FGFR2-BICC1 (F17;B16)*, *FGFR2-CCDC6 (F17;C2)*, *FGFR2-CFAP57 (F17;C12)*, *FGFR2-LRBA (F17;L48)*, *FGFR2-LRRFIP2 (F17;L3)*, *FGFR2-SLMAP (F17;S3)*, *FGFR2-SORBS1 (F17;S3)*, and *FGFR2-ZMYM4 (F17;Z2)* were detected each in one sample. In one case, the kinase domain mutation p.I547V co-existed with the *FGFR2-BICC1 (F17;B3)* translocation. Another three tumors contained known activating *FGFR2* mutations: p.C382R (*n* = 2) and p.F276C (*n* = 1). In addition, a variant of unknown significance, *FGFR2* p.R203S, was found in one case. With the exception of one instance of the *FGFR2* p.C382R mutation, all mentioned *FGFR2* gene alterations were found exclusively in the iCCA samples.

Mutations in the *IDH1* gene, including p.R132C (*n* = 10), p.R132L (*n* = 5), p.R132G (*n* = 4), and p.R132A (*n* = 1), or the *IDH2* gene, including p.R172W (*n* = 2) and p.R172K (*n* = 1), were also found in iCCA cases only. In one case, the *IDH1* p.R132C mutation co-existed with *FGFR2-CFAP57 (F17;C12)* translocation.

*KRAS*, *NRAS*, and *BRAF* mutations were mutually exclusive with *FGFR2* alterations as well as with *IDH1/IDH2* mutations. *KRAS* mutations were more prevalent in eCCAs than in iCCAs or GBAs (Table 2, Fisher’s exact test *p*-value = 0.003). Targetable *BRAF* p.V600E mutations were found in 3/165 (1.8%) BTCs, and *KRAS* p.G12C substitutions were detected in 4/165 (2.4%) BTCs.

Although there are no effective treatment strategies for the BTCs with *PIK3CA* mutations at the moment, these mutations are potentially targetable and, as such, were included in the NGS panel. We found 18/165 (10.9%) of BTCs in this study to carry different *PIK3CA* mutations.

The targeted RNA sequencing panel was used to measure the expression of the *ERBB2*, *FGFR1*, *FGFR2*, *FGFR3*, *FGFR4*, and *PD-L1* genes in a qPCR-like manner: the coverage was determined for one selected fragment within each gene of interest and normalized using referee genes, as described in the Section 2. 

*ERBB2* amplification is a relatively frequent targetable alteration in BTCs [9,10,42]. In this study, 161/168 patients were screened for the presence of *ERBB2* amplifications with PCR, and five positive cases (3.1%) were identified (Table 2). Additionally, two of the studied samples were found to be positive for the *ERBB2* mutations p.S310F and p.R678Q by RNA sequencing. Samples with ERBB2 amplifications or mutations were wild type for the alterations in *FGFR2*, *IDH1/2*, *KRAS*, *NRAS*, or *BRAF* genes. *ERBB2* expression, estimated from NGS data, was found to be significantly increased in all five samples with *ERBB2* gene amplifications (Mann–Whitney *U* test *p*-value = 0.0002) but not in two samples with *ERBB2* point mutations (Figure 2a). One additional wild-type sample had apparently elevated *ERBB2* expression. However, we noticed that in this case the estimated expressions of all the other studied genes were also high. Thus, it is likely that the expression characteristics in this case were subjected to inadequate normalization due to unusual expression levels of the referee genes. 

*FGFR1–4* genes are often subjected to amplifications that are detected across multiple tumor types [43,44], and such events may lead to the overexpression of the respective genes. We found that *FGFR2–4* genes were more frequently overexpressed in iCCA, compared with the eCCA and GBA samples, whereas no such association was observed for the *FGFR1* gene (Figure 2b–e). Among the BTCs that had the highest *FGFR1* gene expression, 8 of 10 cases were positive for mutations in *KRAS*, *NRAS*, *IDH1*, *IDH2*, or *FGFR2* genes. Elevated *FGFR2* gene expression was observed in most cases positive for *FGFR2*-translocations or *FGFR2* mutations (Figure 2b). 

*PD-L1* expression is routinely measured by immunohistochemistry (IHC) to predict the tumor response to PD-L1 inhibitors in particular tumor types, such as lung cancer, but the significance of this marker in BTCs is contradictory [4]. Interestingly, two of five BTCs with *ERBB2* amplifications in this study were among the samples with the highest *PD-L1* expression levels (Figure 2f). Recent reports have suggested that high PD-L1 expression may contribute to acquired tumor resistance to trastuzumab [45,46]. Patients participating in our study did not receive HER2-targeted therapy; therefore, we cannot comment on the potential interference between the status of these two genes. Tumors with co-incident upregulation of *ERBB2* and *PD-L1* may deserve particular attention in future investigations. 

### 3.3. Analysis of a Subset of iCCA Cases with Illumina TruSight Tumor 170 Kit: Comparison of the Results Obtained with the Two Methods

The Illumina TruSight Tumor 170 kit is a widely applied and well-recognized hybridization-based panel that allows for the detection of translocations (in RNA) and mutations (in DNA) involving 170 cancer-associated genes. With this kit, we prepared RNA and DNA libraries from the 47 iCCA samples previously analyzed with the newly developed 3′ RACE-based targeted RNA sequencing approach. Samples for the analysis with Illumina TruSight Tumor 170 kit were selected primarily based on the availability of the additional FFPE tissue, as most clinical iCCA samples in this study were limited biopsy specimens. 

More than 10 million reads that passed filters were obtained for each of the sequenced DNA and RNA libraries. The median template length in the RNA libraries (121.5 bp, range 80–150) was lower than in the DNA libraries (154.5 bp, range 112–191). In all but two DNA libraries more than 95% of exons had at least 100X coverage. 

There was a good concordance between the results obtained with the 3′ RACE-based targeted RNA panel and Illumina TruSight Tumor 170 kit (Appendix A). Both methods successfully identified 9/9 (100%) *FGFR2* translocation-positive cases. The results of mutational analysis for the gene regions included in both panels was concordant in 43/47 cases (91.5%). The discrepant cases included two mutations (*IDH1* p.R132L and *KRAS* p.Q61H) in two samples that were missing in the Illumina TruSight Tumor 170 results: both variants were present in the raw data but were filtered out during bioinformatic analysis. *KRAS* p.Q61H was missed in one of the two samples with low sequencing coverage because the number of reads was insufficient for variant calling. In case of *IDH1* p.R132L mutation, the reason seemed to be the low fraction (2%) of the mutant allele in the DNA sample. In the RNA sample, however, the prevalence of the mutant allele was higher (6.7%) and that helped to identify mutations with the 3′ RACE-based targeted RNA sequencing approach. Notably, the mutant allele fraction was higher in RNA sequencing results compared with the DNA sequencing results for most of the known activating somatic mutations in all samples (Appendix A). In another two cases, variants of unknown significance in the *FGFR2* and *FGFR3* genes were found by DNA sequencing but were absent in the 3′ RACE-based RNA sequencing results. These variants, perhaps, are “passenger” mutations and are therefore not necessarily expressed in the tumor cells. RNA sequencing can only identify mutations in the expressed genes, and this could be the reason for the failure to identify these particular variants.

### 3.4. Other Genetic Alterations Detected in iCCA

Analysis of the data obtained with the Illumina TruSight Tumor 170 panel revealed many additional variants in the studied iCCA samples. In accordance with previously published data [9], most frequently affected by mutations were the *BAP1* (10/47 cases, 21.3%), *TP53* (9/47 cases, 19.1%), and *ARID1A* (7/47, 14.9%) genes (Appendix A). Mutations in genes involved in DNA double-strand break repair were also detected in a significant proportion of the iCCA cases. *BRCA2* alterations were observed in four cases: two mutations listed as “Pathogenic” and “Pathogenic/Likely_pathogenic” in the ClinVar database [47] (*BRCA2* p.R2336H and *BRCA2* p.H2623R) and one novel variant (*BRCA2* p.A1170S) had high mutant allele fractions, which could reflect their germline origin. Another frameshift mutation in *BRCA2* was found in MSI-positive sample, with only 10.8% mutant allele prevalence. Frameshift mutations in *ABRAXAS1*, *MRE11*, and *PALB2* were found in one case each, and more variants of unknown significance were identified in some other genes involved in the homologous recombination and double-strand break repair (Appendix A). Two iCCA cases with known pathogenic variants in the *MLH1* gene were identified. However, the standard pentaplex panel identified MSI in only one of these cases. Interestingly, the MSI-positive tumor also contained *ERCC2* mutation with 50% prevalence of the altered allele. Amplifications of *MYC* (four cases), *MDM2* (two cases), *MDM4* (two cases), and several other genes were identified with the Illumina TruSight Tumor 170 panel in a subset of tumors (Appendix A). Surprisingly, *ESR1-CCDC170* translocation was found in one tumor, which is, to our knowledge, the first case of *ESR1*-fusion-positive iCAA described to date.

Translocations involving known targetable genes, such as *ALK*, *ROS1*, *RET*, and *NTRK1–3,* were previously reported as rare targetable alterations in BTCs (mainly, in iCCA cases) [48,49,50,51,52]. We applied a panel of RT-PCR-based tests [28,29,30,31] to the iCCA cases that were not subjected to the analysis with Illumina TruSight Tumor 170 panel; however, no such rearrangements or the *MET* exon 14 skipping mutations, were found.

## 4. Discussion

The molecular profiling of BTCs, and particularly of iCCA, is challenging because a wide spectrum of targetable alterations is characteristic for these tumors. Here, we developed a targeted RNA sequencing panel for the analysis of main clinically important alterations in BTCs on the basis of a well-known 3′ RACE technology. The technique 3′ RACE allows for detection of translocations with unknown gene partners located at the 3′ end of the chimeric transcript. The suggested targeted RNA library preparation pipeline is fast and inexpensive, even in comparison with other anchored multiplex PCR-based approaches. In fact, it is probably the simplest and least expensive method of RNA library preparation that allows for the identification of translocations with unknown gene partners. It lacks the adapter ligation step, and no modified oligonucleotides are needed. Targeted RNA libraries for NGS sequencing can be prepared with the use of only two enzymes, the M-MuLV reverse transcriptase and the hot-start Taq polymerase, which are most widely applied in molecular genetics studies. Importantly, we developed a framework that can facilitate the design of the anchored multiplex PCR-based NGS panels in the future (Appendix A). 

For the analysis of data obtained with the 3′ RACE-based NGS approach, we developed a bioinformatics pipeline that allowed us to analyze translocations, mutations, and the expression of the genes included in the panel. The analysis of translocations was successfully carried out with the well-known STAR-Fusion package [33]. However, we found that the existing freely available bioinformatics software tools have insufficient sensitivity when the mutational calling is applied to tumor-only RNA sequencing data. Thus, we developed new scripts on the basis of the *pysam* python module [34] that are more suitable for the analysis of data obtained using 3′ or 5′ RACE-based NGS. The new bioinformatics pipeline was successfully applied for the analysis of clinical BTC samples in this study; however, it may need additional testing and, possibly, further improvement before it can be recommended for wider application. 

NGS allows for the simultaneous analysis of multiple molecular markers. The developed NGS panel is specifically designed for the analysis of clinically important alterations characteristic of BTCs. The use of such a panel in clinical practice would allow more BTC patients to benefit from more efficient and less toxic treatment options. 

The study of 168 clinical BTC samples demonstrated the ability of the developed approach to identify most relevant targetable alterations characteristic for these tumors, including *FGFR2* translocations and mutations, *IDH1* mutations, *KRAS* and *BRAF* mutations, *ERBB2* amplifications (determined by the measurement of gene overexpression), and *ERBB2* mutations. The frequencies of genetic alterations identified in our sample are in good agreement with the previously reported data. The analysis of gene fusions is particularly complicated due to various technical limitations. The reported frequency of *FGFR2* gene rearrangements in iCCA usually falls within 13–15% [9,53,54,55]. The current study revealed *FGFR2* translocations in 14/82 (17%) iCCA cases, which seems an indicator of the good reliability of our NGS pipeline. Boscoe et al. [56] systematically reviewed 45 studies and calculated that the overall frequency of *IDH1* mutations in iCCA is 13.1%; however, it was higher in non-Asian centers than in Asian centers (16.6% and 8.8%, respectively) [56]. Notably, both *FGFR2* fusions and *IDH1* mutations are rare in fluke-associated cholangiocarcinomas [10,57]. This can explain, at least partly, the observed interstudy variations. The increased frequency of *FGFR2* translocations and *IDH1* mutations observed in the current data set is likely to be attributed to a relatively low prevalence of fluke-associated CCAs in Russia. In the current study, *IDH1* mutations were observed in 24% of the analyzed iCCA samples. Both *FGFR2* fusions and *IDH1* mutations are known to be almost exclusively found in iCCAs but not in other categories of BTCs [53,54,55,56]. In accordance with that, no instances of *FGFR2* fusion or *IDH1* mutation were found in eCCA or GBA samples in our study. The latter fact suggests that our diagnostic assay does not produce false-positive results.

We observed a higher frequency of *KRAS* mutations in eCCAs than iCCAs and GBAs (Table 2). An increased frequency of *KRAS* mutations in eCCAs was also reported in other studies [55,58]. Targetable *BRAF* p.V600E mutations are rare in BTCs, with the highest prevalence (1–2%) in the iCCA subtype [9,59,60]. In the current study, *BRAF* p.V600E mutations were found in 2/80 (2.4%) iCCA samples.

*ERBB2* amplifications are infrequent in fluke-negative CCAs [10]. According to the literature data, a high prevalence of *ERBB2* amplifications (up to 20%) is characteristic of GBAs [61]. However, we observed a lower *ERBB2* amplification rate (7.3%) in the studied GBA samples. It should be kept in mind that the measurement of gene expression level or copy number variations by conventional NGS or real-time PCR may be compromised by low tumor cell content or high intratumoral heterogeneity. In such cases, IHC and/or FISH can provide better sensitivity by allowing visualization of individual tumor cells.

The Illumina TruSight Tumor 170 kit proved to be an excellent instrument for the detailed study of 170 cancer-associated genes, which included the analysis of mutations, translocations, and copy number alterations. The parallel analysis of a subset of iCCA tumors (n = 47) with the Illumina TruSight Tumor 170 kit demonstrated nearly full concordance for the status of genomic regions that were covered by both panels. There were two cases in which mutations identified with the 3′ RACE-based technology were missed by the Illumina TruSight Tumor 170 method; however, the respective variants were present in the raw NGS reads and were not called due to the insufficient coverage in one sample and the insufficient mutant allele fraction in the other tumor. On the other hand, there were two non-hotspot variants that were detected in the DNA using the Illumina TruSight Tumor 170 kit but not in the RNA of the same samples using the 3′ RACE-based NGS. Failure of RNA-based NGS to detect these mutations is likely to attributed to the lack of expression of non-functional variant gene copies.

The results obtained by parallel analysis of mutations using the DNA-based method (Illumina TruSight 170 kit) and RNA-based approach (3′ RACE-based NGS) highlight the apparent advantage of RNA-based technologies in studying the activating mutations in oncogenes. The mutated allele is often overexpressed by tumor cells and can be detected by the analysis of RNA, even in a sample with low tumor content. At the same time, “passenger” mutations can be missed by RNA sequencing due to lack of expression of the corresponding alleles. 

The developed method has certain limitations. As with any RNA-based approach, it can only detect activating mutations in oncogenes, whereas some clinically significant alterations in BTCs may affect tumor suppressor genes or DNA repair genes. In particular, known pathogenic mutations in the *BRCA2* gene were found in two iCCA samples that were analyzed with the Illumina TruSight Tumor 170 kit. Recently, it was suggested that BTCs with mutations in *BRCA1/2* genes may benefit from therapy with poly(ADP-ribose) polymerase inhibitors (PARPi) [62,63]. Pathogenic *MLH1* mutations were observed in another two iCCA cases; interestingly, only one of these cases was MSI-positive, as determined by the pentaplex PCR panel. Inactivation of DNA mismatch repair genes may not necessarily result in MSI, as tumors with a low cell proliferation rate sometimes do not accumulate an excessive number of microsatellite shifts [64].

The important limitation of the 3′ RACE-based NGS is its inability to detect translocations with unknown partner genes located at the 5′ end of the chimeric transcript. These types of translocations include targetable *ALK, ROS1, RET,* and *NTRK1-3* genes fusions, which were detected in a small fraction of BTCs, primarily iCCAs, in some studies [48,49,50,51,52]. We assessed these translocations in available iCCA samples using either the Illumina TruSight Tumor 170 kit or the previously described PCR-based tests, including tests for unbalanced expression of the 5′/3′ mRNA ends [29,30,31], but found no instances of such alterations. 

In conclusion, the method of targeted RNA sequencing presented here, which was developed on the basis of 3′ RACE technology, demonstrated good performance in the analysis of clinically relevant genetic alterations in BTCs. The proposed technology has good potential for various applications in diagnostic and research settings, as it is characterized by simplicity, low cost, and the ability to simultaneously analyze a spectrum of targets represented by oncogenic translocations, activating driver mutations, and overexpression of a number of clinically significant genes. 

## Figures and Tables

**Figure 1 diagnostics-13-03168-f001:**
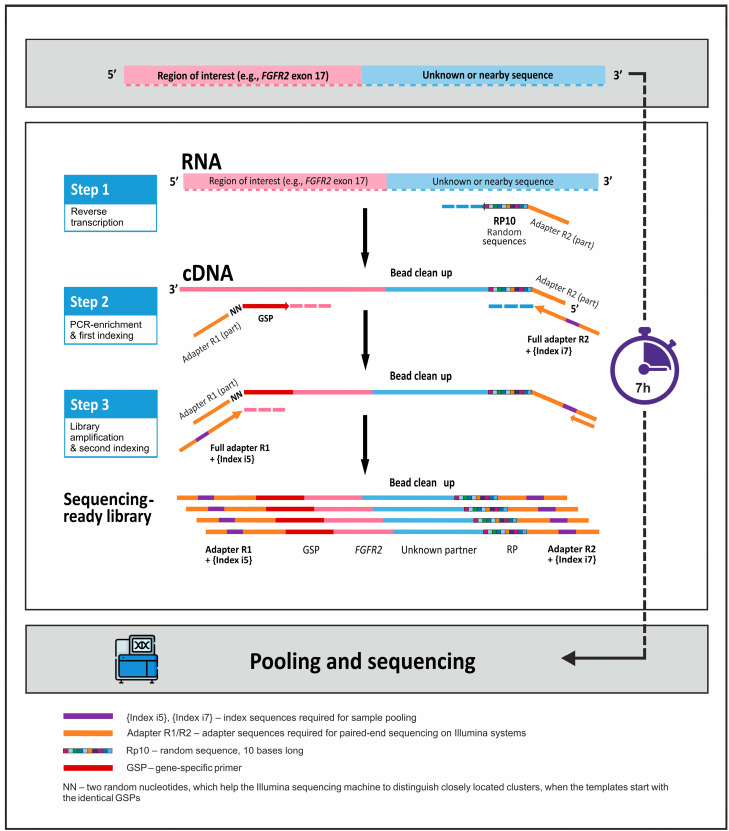
NGS library preparation using the 3′ RACE-based approach.

**Figure 2 diagnostics-13-03168-f002:**
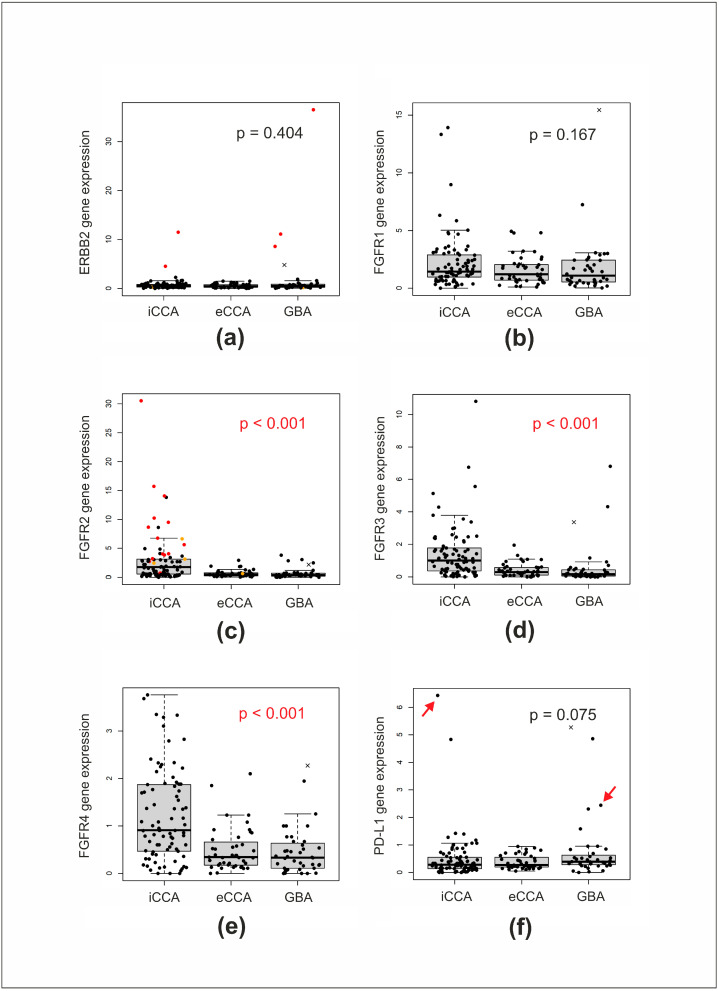
Analysis of mRNA expression of (**a**) *ERBB2*, (**b**) *FGFR1*, (**c**) *FGFR2*, (**d**) *FGFR3*, (**e**) *FGFR4*, and (**f**) *PD-L1* genes in biliary tract cancers: intrahepatic cholangiocarcinomas (iCCA), extrahepatic cholangiocarcinomas (eCCA), and gallbladder adenocarcinomas (GBA). Normalized expression values for each sample are shown as dots on the graphs. The Kruskal–Wallis test *p*-values for the comparison of the three groups of tumors are given. Red and orange dots in (**a**) indicate samples with *ERBB2* gene translocations and mutations, respectively. “x” indicates a sample with apparently failed normalization. Red and orange dots in (**c**) represent samples with *FGFR2* translocations and mutations, respectively. Arrows in (**f**) indicate high *PD-L1* expression levels in two of the five samples with ERBB2 amplifications. iCCA—intrahepatic cholangiocarcinoma; eCCA—extrahepatic cholangiocarcinoma; GBA—gallbladder adenocarcinoma.

**Table 1 diagnostics-13-03168-t001:** The composition of the targeted RNA sequencing panel developed for the study of BTCs.

Genes	Detected Alterations	Number of Primers
*FGFR1–4*	translocations, mutations in hot-spot regions, overexpression (as a marker of gene amplification)	29
*IDH1/2*	*IDH1* p.R132, *IDH2* p.R140 and *IDH2* p.R172 hot-spot mutations	3
*ERBB2*	overexpression (as a marker of the gene amplification), mutations in the hot-spot regions	9
*KRAS/NRAS*	mutations in the hot-spot regions (exons 2–4 for *KRAS* and exons 2–3 for *NRAS*)	7
*BRAF*	mutations in exon 15	1
*PIK3CA*	mutations in the hot-spot regions	17
*PD-L1*	expression	1
*DDX23, GOLGA5, SEL1L*	referee genes, required for the analysis of the expression of selected genes	3

**Table 2 diagnostics-13-03168-t002:** Main clinical characteristics and results of molecular analyses.

	iCCA (*n* = 83)	eCCA (*n* = 44)	GBA (*n* = 41)	*p*-Value
Age				
median (range)	61 (18–83)	63 (24–75)	66 (42–91)	0.007 ^1^
Gender				
% female	60.2	63.6	73.2	0.366 ^2^
MSI status				
MSI+	2 (2.6%)	0 (0%)	1 (2.6%)	0.616 ^3^
MSI−	76 (97.1%)	41 (100%)	38 (97.4%)	
nd	4	3	2	
*ERBB2* amplification				
present	2 (2.6%)	0 (0%)	3 (7.3%)	0.118 ^3^
absent	74 (97.4%)	44 (100%)	38 (92.7%)	
nd	7	0	0	
*FGFR2* gene status				
fusion or mutation	17 (20.7%)	1 (2.3%)	0 (0%)	<0.001 ^3^
wild type	65 (79.3%)	42 (97.7%)	40 (100%)	
nd	1	1	1	
*IDH1/2* mutations				
present	23 (28%)	0 (0%)	0 (0%)	<0.001 ^3^
absent	59 (72%)	43 (100%)	40 (100%)	
nd	1	1	1	
*KRAS* mutation				
present	11 (13.4%)	16 (37.2%)	4 (10%)	0.003 ^3^
absent	71 (86.6%)	27 (62.8%)	36 (90%)	
nd	1	1	1	
*NRAS* mutation				
present	3 (3.7%)	0 (0%)	1 (2.5%)	0.680 ^3^
absent	79 (96.3%)	43 (100%)	39 (97.5%)	
nd	1	1	1	
*BRAF* mutation				
present	2 (2.4%)	0 (0%)	1 (2.5%)	0.614 ^3^
absent	80 (97.6%)	43 (100%)	39 (97.5%)	
nd	1	1	1	
*ERBB2* mutation				
present	1 (1.2%)	0 (0%)	1 (2.5%)	0.494 ^3^
absent	81 (98.8%)	43 (100%)	39 (97.5%)	
nd	1	1	1	
*PIK3CA* mutation				
present	9 (11%)	6 (14%)	3 (7.5%)	0.722 ^3^
absent	73 (89%)	37 (86%)	37 (92.5%)	
nd	1	1	1	

Abbreviations: iCCA—intrahepatic cholangiocarcinoma; eCCA—extrahepatic cholangiocarcinoma; GBA—gallbladder adenocarcinoma; nd—not determined. ^1^ Kruskal–Wallis test *p*-value. ^2^ Chi-squared test *p*-value. ^3^ Fisher’s exact test *p*-value.

## Data Availability

The data that support the findings of this study are available from the corresponding author upon reasonable request.

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
