# Peer review of "Molecular Analysis of Biliary Tract Cancers with the Custom 3′ RACE-Based NGS Panel"

_diagnostics, 2023, doi:10.3390/diagnostics13203168_

Round 1
Reviewer 1 Report
Manuscript that involves innovative technology which leads to the verification of results, as mentioned by the author, the 3'RACE technology seems to be very useful considering its limitations in the identification of non-hot spots in translocations, in the text it is advisable to be more concise in the methodology, which is a bit repetitive; You should emphasize in the discussion the possible bias of the analysis of results when using your own bioinformatics pipeline. Clarify in line 417 what is meant by the term “almost agreement of the tumor kit 170 of the status of the genomic regions”. It is suggested to unify the units of microliters with the mu symbol of the Greek system. The conclusions as presented seem to detract from the exhaustive work, it is suggested to highlight the panel designed with 3'RACE technology. NGS together with 3'RACE leads to the new technologies proposed in the not too distant future
Understandable and acceptable English, only very technical language that is understandable to use
Author Response
Comment: In the text it is advisable to be more concise in the methodology, which is a bit repetitive.
Response: In this work we presented a new method and provided a full description of all the procedures. We believe that if someone decides to use this method in his or her own study, all provided details are likely to be important. Thus, we request to maintain full methodological content in the manuscript, despite that it may seem somewhat lengthy.
Comment: You should emphasize in the discussion the possible bias of the analysis of results when using your own bioinformatics pipeline.
Response: We have now addressed this issue in the article:
“For the analysis of data obtained with the 3’ RACE-based NGS approach we developed a bioinformatics pipeline, which allowed us to analyze translocations, mutations and expression of the genes included in the panel. The analysis of translocations was successfully carried out with the well-known STAR-Fusion package [33]. However, we found that the existing freely available bioinformatics software tools have insufficient sensitivity when the mutational calling is applied to the tumor-only RNA sequencing data. Thus, we developed the new scripts on the basis of the pysam python module [34], which is more suitable for the analysis of data obtained using 3’ or 5’ RACE-based NGS. The new bioinformatics pipeline was successfully applied for the analysis of clinical BTC samples in this study; however, it may need additional testing and, possibly, further improvement, before it can be recommended for the wider application.”
Comment: Clarify in line 417 what is meant by the term “almost agreement of the tumor kit 170 of the status of the genomic regions.
Response: We added more detailed explanation of this message in the Discussion:
“The parallel analysis of a subset of iCCA tumors (n = 47) with the Illumina TruSight Tumor 170 kit demonstrated nearly full concordance for the status of genomic regions, which were covered by both panels. There were two cases, in which mutations identified with the 3’ RACE-based technology were missed by the Illumina TruSight Tumor 170 method, however the respective variants were present in the raw NGS reads and were not called due to the insufficient coverage in one sample and the insufficient mutant allele fraction in the other tumor. On the other hand, there were two non-hotspot variants, which were detected in the DNA using the Illumina TruSight Tumor 170 kit, but not in RNA of the same samples using the 3’ RACE-based NGS. Failure of RNA-based NGS to detect these mutations is likely to attributed to the lack of expression of non-functional variant gene copies.”
Comment: It is suggested to unify the units of microliters with the mu symbol of the Greek system.
Response: This is done.
Comment: The conclusions as presented seem to detract from the exhaustive work, it is suggested to highlight the panel designed with 3'RACE technology.
Response: The conclusion was reformulated as follows:
“In conclusion, the presented here method of targeted RNA sequencing, which has been developed on the basis of 3’ RACE technology, demonstrated good performance in the analysis of clinically relevant genetic alterations in BTCs. The proposed technology has a good potential for various applications in diagnostic and research settings, as it is characterized by simplicity, low-cost, and the ability to simultaneously analyze a spectrum of targets represented by oncogenic translocations, activating driver mutations and gene overexpression.”
Reviewer 2 Report
The authors developed a targeted RNA sequencing panel for the detection of main gene alterations in biliary tract cancers (BTCs) on the basis of 3' rapid amplification of cDNA ends (3' RACE) technology, and applied it for the analysis of clinical samples. It is an innovative sequencing technology that deeply explore the internal mechanisms and therapeutic targets of BTCs. The study shows promising, but I believe addressing the mentioned issues will significantly enhance the quality and clarity of the manuscript:
1. In Introduction part, more literature support is needed about the advantages of anchored multiplex PCR methods in tumor research, more researches are needed to supplement to demonstrate the importance of 3’ RACE-based RNA sequencing.
2. Please provide a better discussion on the meaning of clinical translation in this targeted RNA sequencing panel.
3. Please provide a better discussion on the usefulness of Illumina TruSight Tumor 170 kit, advantages and disadvantages in Illumina TruSight Tumor 170 kit and 3’RACE-based targeted RNA panel.
4. What is the correlation between PD-L1 expression and ERBB2 amplification in BTCs, what is the meaning expressed in Figure 2f?
5. The manuscript briefly mentioned translocations with the partner genes located at the 5’ end of the chimeric transcript, but it would greatly benefit from a more comprehensive explanation to ensure clarity for readers.
6. Compared with traditional NGS sequencing methods, how is the cost-effectiveness of custom 3’RACE-based NGS panel?
7. It is recommended to compare the results with the reported gene site mutations and frequencies of BTCs.
Author Response
Comment: In Introduction part, more literature support is needed about the advantages of anchored multiplex PCR methods in tumor research, more researches are needed to supplement to demonstrate the importance of 3’ RACE-based RNA sequencing.
Response: The following text was added to the Introduction:
“Anchored multiplex PCR methods are able to detect translocation with unknown partner genes, but only if the breakpoint positions in at least one of the genes participating in the translocation are well-defined [17]. They are generally more rapid and less expensive when compared to hybridization-based NGS library preparation protocols. PCR enrichment takes only about 2 hours, while the hybridization requires overnight incubation. Anchored multiplex PCR panels usually focus on a relatively narrow number of genomic segments and, therefore, they are smaller than hybridization-based NGS tests. Also, fewer NGS reads are required to achieve sufficient coverage in each region of interest, and more samples can be multiplexed for each run when PCR enrichment is utilized. Most commonly used anchored multiplex PCR methods require the RNA to be converted to the double-strand cDNA, followed by the adapter ligation, PCR-enrichment and final amplification of the libraries [17]. On the basis of this approach, two popular commercial NGS solutions were developed, namely Archer FusionPlex (ArcherDX, USA) and QIAseq RNAscan (Qiagen, Germany). Multiple studies relied on the use of these kits, thus supporting the feasibility of the anchored multiplex PCR method [21-24]. Alternatively, the adapter sequence can be attached to the cDNA molecules by the reverse transcriptase; this procedure is utilized in the template-switch-based 5’ rapid amplification of cDNA ends (5’ RACE) or the 3' rapid amplification of cDNA ends (3' RACE) protocols [25,26]. The 5’ RACE can be used when all the known translocation partners are located at the 3’ part of the chimeric transcripts, while the respective unknown partners are located at the 5’ part of the chimeric transcripts. On the contrary, the 3’ RACE can be utilized only if all the unknown translocation partners are expected to occupy the 3’-ends of the mRNA. Importantly, the 3’ RACE and the 5’ RACE-based approaches allows for simple and straightforward in-house design of the custom RNA NGS tests.”
Comment: Please provide a better discussion on the meaning of clinical translation in this targeted RNA sequencing panel.
Response: The Discussion was extended with the following text:
“NGS allows for simultaneous analysis of multiple molecular markers. The developed NGS panel is specifically designed for the analysis of clinically important alterations characteristic of BTCs. The use of such panel in clinical practice would allow more BTC patients to benefit from more efficient and less toxic treatment options.”
Comment: Please provide a better discussion on the usefulness of Illumina TruSight Tumor 170 kit, advantages and disadvantages in Illumina TruSight Tumor 170 kit and 3’RACE-based targeted RNA panel/
Response: The following was added to the Discussion:
“Illumina TruSight Tumor 170 kit proved to be an excellent instrument for the detailed study of 170 cancer-associated genes, which includes the analysis of mutations, translocations and copy number alterations.”
“The results obtained by parallel analysis of mutations using DNA-based method (Illumina TruSight 170 kit) and RNA-based approach (3’ RACE-based NGS) highlight the apparent advantage of RNA-based technologies in studying the activating mutations in oncogenes. The mutated allele is often overexpressed by tumor cells and can be detected by the analysis of RNA even in a sample with low tumor content. At the same time, “passenger” mutations can be missed by RNA sequencing due to lack of expression of the corresponding alleles.”
Comment: What is the correlation between PD-L1 expression and ERBB2 amplification in BTCs, what is the meaning expressed in Figure 2f?
Response: There is no clear correlation between the two; in fact, it can be a coincidence. However, we have found it intriguing that among the five samples with ERBB2 amplifications two also had the unusually high PD-L1 expression. The following considerations were added to the text:
“Recent reports suggested that high PD-L1 expression may contribute to acquired tumor resistance to trastuzumab [45,46]. Patients participating in our study did not receive HER2-targeted therapy, therefore we cannot comment on the potential interference between the status of these two genes. Tumors with co-incident upregulation of ERBB2 and PD-L1 may deserve particular attention in future investigations.”
Comment: The manuscript briefly mentioned translocations with the partner genes located at the 5’ end of the chimeric transcript, but it would greatly benefit from a more comprehensive explanation to ensure clarity for readers.
Response: We provide comprehensive list of partner genes:
“Translocations involving FGFR2 gene were detected in 14 samples: FGFR2-AHCYL1 (F17;A2), FGFR2-BICC1 (F17;B3) and FGFR2-VCL (F17;V2) variants were found in two cases each, while FGFR2-BICC1 (F17;B16), FGFR2-CCDC6 (F17;C2), FGFR2-CFAP57 (F17;C12), FGFR2-LRBA (F17;L48), FGFR2-LRRFIP2 (F17;L3), FGFR2-SLMAP (F17;S3), FGFR2-SORBS1 (F17;S3) and FGFR2-ZMYM4 (F17;Z2) were detected each in one sample.”
Comment: Compared with traditional NGS sequencing methods, how is the cost-effectiveness of custom 3’RACE-based NGS panel?
Response: The following phrase was added to the first paragraph of the Discussion to clarify the issue:
“In fact, it is probably the simplest and least expensive method of RNA library preparation, which allows for the identification of translocations with unknown gene partners.”
Comment: It is recommended to compare the results with the reported gene site mutations and frequencies of BTCs.
Response: The discussion on mutation frequencies was added:
“The frequencies of genetic alterations identified in our sample are in good agreement with the previously reported data. The analysis of gene fusions is particularly complicated due to various technical limitations. The reported frequency of FGFR2 gene rearrangements in iCCA usually falls within 13-15% [9, 53-55]. Current study revealed FGFR2 translocations in 14/82 (17%) iCCA cases, which seems an indicator of good reliability of our NGS pipeline. Boscoe et al. [56] systematically reviewed 45 studies and calculated that the overall frequency of IDH1 mutations in iCCA is 13.1%; however it was higher in non-Asian than in Asian centers (16.6% and 8.8%, respectively) [56]. Notably, both FGFR2 fusions and IDH1 mutations are rare in fluke-associated cholangiocarcinomas [10, 57]. This can explain, at least partly, the observed interstudy variations. The increased frequency of FGFR2 translocations and IDH1 mutations observed in the current data set is likely to be attributed to a relatively low prevalence of fluke-associated CCAs in Russia. In the current study, IDH1 mutations were observed in 24% of the analyzed iCCA samples. Both FGFR2 fusions and IDH1 mutations are known to be almost exclusively found in iCCAs, but not in other categories of BTCs [53-56]. In accordance with that, no instances of FGFR2 fusion or IDH1 mutation were found in eCCA or GBA samples in our study. The latter fact suggests that our diagnostic assay does not produce false-positive results.
We observed higher frequency of KRAS mutations in eCCAs as compared to iCCAs and GBAs (Table 2). Increased frequency of KRAS mutations in eCCAs was also reported in other studies [55,58]. Targetable BRAF p.V600E mutations are rare in BTCs, with the highest prevalence (1-2%) in the iCCA subtype [9,59,60]. In the current study, BRAF p.V600E mutations were found in 2/80 (2.4%) iCCA samples.
ERBB2 amplifications are infrequent in the fluke-negative CCAs [10]. High prevalence of ERBB2 amplifications (up to 20%) is characteristic of GBAs, according to literature data [61]. However, we observed lower ERBB2 amplification rate (7.3%) in the studied GBA samples. It should be kept in mind that the measurement of gene expression level or copy number variations by conventional NGS or real-time PCR may be compromised by low tumor cell content or high intratumor heterogeneity. In such cases, IHC and/or FISH can provide better sensitivity by allowing visualization of individual tumor cells.”